# An RPA-CRISPR/Cas12a-based rapid and sensitive nucleic acid method for detection of *Toxoplasma gondii* in tissue and blood samples

Yilin Wang,[1,2,3] Ziyang Qin,[1,2,3] Qinglin Wang,[1,2,3] Yurong Yang,[1] Chunhao Gu,[1,2,3] Fuchang Yu,[1,2,3,4] Yayun Wu,[1,2,3] Long xian Zhang[1,2,3]

**ABSTRACT**   *Toxoplasma gondii* is a zoonotic pathogen that can infect humans and a wide range of warm-blooded animals, posing a significant threat to human health and the livestock industry. The development of a time-saving, highly sensitive, and specific method for the detection of *T. gondii* in tissue and blood samples is crucial to the monitoring, prevention, and control of toxoplasmosis. In this study, we evaluated the efficiency of a previously described method, termed REPORT, that integrates recombinase polymerase amplification with CRISPR/Cas12a for the detection of *T. gondii* nucleic acids. We evaluated the limit of detection (LOD) and specificity of the extended REPORT method using prepared target DNA in addition to tissue and blood samples. Furthermore, we validated the accuracy of *T. gondii* detection in clinical samples using the REPORT-based method in comparison with nested PCR based on the B1 gene. Sensitivity tests showed that the LOD of the REPORT-based fluorescence method and the lateral flow strip method were 3.7 copies /μL for target DNA, 3.1 tachyzoites/g for tissue samples, and five tachyzoites/mL for blood samples. Specificity tests suggested that the REPORT method had good specificity and did not cross-react with several common parasites. The method performed well for clinical DNA samples, demonstrating its ability for use in on-site detection.

**IMPORTANCE**   *Toxoplasma gondii* can infect over 200 species of warm-blooded animals, including humans, posing not only a significant threat to public health systems but also causing substantial economic losses to the global livestock industry. Current diagnostic methods are slow, equipment-dependent, and impractical for field use. This study addresses these limitations by developing REPORT, a rapid, ultrasensitive nucleic acid test combining recombinase polymerase amplification and CRISPR/Cas12a. The REPORT detects *T. gondii* in tissue and blood samples within 1 h at low cost, requiring only a portable heater. Its visual results (fluorescence or test strips) enable on-site use without specialized training, achieving 100% accuracy versus nested PCR. With a sensitivity of 3.1 parasites per gram of tissue and five parasites per milliliter of blood, this method revolutionizes toxoplasmosis screening in resource-limited clinics, farms, and food safety inspections, empowering timely interventions to curb transmission and improve public health outcomes.

**KEYWORDS**   *Toxoplasma gondii*, visualized detection, recombinase polymerase amplification, CRISPR/Cas12a, On-site detection

*T*oxoplasma gondii (*T. gondii*), which is an obligate intracellular parasite belonging to the Apicomplexa phylum, can infect a wide range of warm-blooded hosts, including humans, birds, and rodents (1). As the sole definitive host of *T. gondii*, felids can shed oocysts that are highly resistant to environmental conditions (2). It has been estimated

**Peer Reviewer** Saleem Khteer Al-hadraawy, University of Kufa, Najaf, Iraq

Address correspondence to Long xian Zhang, zhanglx8999@henau.edu.cn.

The authors declare no conflict of interest.

See the funding table on p. 10.

that a third of the world's population has been infected. Toxoplasmosis is generally contracted by ingesting undercooked or raw meat containing viable tissue cysts or by ingesting food or water contaminated with oocysts (3, 4). In immunocompetent individuals, *T. gondii*-induced infections typically present as subclinical or asymptomatic. However, in immunocompromised individuals, the infection can lead to life-threatening conditions such as toxoplasmic encephalitis, myocarditis, and pneumonitis. Infection occurring during pregnancy may result in significant harm to the fetus, including long-term disabling sequelae, stillbirth, and fetal death (5, 6). Owing to its characteristic latent infection and widespread transmission, *T. gondii* infection has become one of the most significant challenges to global public health (7). Therefore, the development of rapid, sensitive, and specific detection methods for *T. gondii* is crucial for ensuring food safety and preventing and controlling toxoplasmosis in animals.

The diagnosis of *T. gondii* in animals can be carried out using a range of techniques, including etiological, immunological, and molecular methods (8). The identification of *T. gondii* in animal blood and tissue samples has traditionally relied on microscopic examination. Nevertheless, using light microscopy for this purpose not only is time-consuming but also requires experienced and skilled professional technicians to ensure accurate detection results (9). The isolation of *T. gondii* through bioassays using laboratory animals is usually considered the gold standard for detecting infection. However, bioassays are expensive and time-consuming (usually requiring 6 weeks), a situation that limits their feasibility for large-scale screening (10, 11). Serological methods are currently used for the diagnosis of *T. gondii* in animal (12). To identify various antibody classes or antigens, a variety of serological tests, including dye tests, modified agglutination tests, enzyme-linked immunosorbent assays, indirect fluorescent antibody tests, and indirect hemagglutination assays, have been developed. Although these methods are simple and rapid, they may fail to detect infections during the window period in animals and can be affected by interfering factors such as rheumatoid elements (8, 13). Polymerase chain reaction (PCR) and quantitative real-time PCR (qPCR) assays have become increasingly important in the molecular detection of *T. gondii* (14, 15). Nevertheless, they rely on thermal cycler, which restricts their application in point-of-care (POC) testing on farms for rapid screening in resource-limited veterinary field settings or slaughterhouses. In recent years, isothermal amplification techniques have emerged such as loop-mediated isothermal amplification (LAMP) and recombinase polymerase amplification (RPA), offering the potential for developing on-site detection tools for animal *T. gondii* diagnosis (16, 17). While LAMP has several advantages, the design of its primers is complex, and the method is susceptible to aerosol contamination, potentially leading to false positive results (18). RPA operates at temperatures between 37°C and 42°C and completes the process within 30 min. The method can detect extremely low levels of pathogen-specific nucleic acids without requiring sophisticated laboratory equipment, making it highly suitable for rapid on-site detection of *T. gondii* in animal blood and tissue samples (19).

Prokaryotic organisms, including bacteria and archaea, have an immune and/or antiviral mechanism that involves clustered regularly interspaced short palindromic repeats (CRISPR), accompanied by a CRISPR-associated protein (Cas) and directed by a short CRISPR RNA (crRNA) (20). A number of studies have shown that Cas12a, a component of the class 2 Type V CRISPR-Cas system, exhibits collateral cleavage activity toward non-targeted ssDNAs after the establishment of the ternary Cas12a/crRNA/target DNA complex (21). The CRISPR/Cas12a-based detection system has been extensively utilized in the detection of pathogens such as *Cryptosporidium parvum* (22), African swine fever virus (23), and *Staphylococcus aureus* (24). The method is simple to operate, has high sensitivity and specificity, and does not rely on precision instruments (25).

In a previous study, we combined recombinase polymerase amplification with the CRISPR/Cas12a *trans*-cleavage system to establish an end-point diagnostic to rapidly detect *Giardia duodenalis* (*G. duodenalis*) assemblages A and B nucleic acids. We termed this method REPORT. The results were observed either as fluorescence under UV light or

using a lateral flow strip (LFS) biosensor (26). In the present study, we aimed to expand the application of the REPORT and establish an end-point nucleic acid detection method for *T. gondii*.

## MATERIALS AND METHODS

### Sample information

The DNA of *Neospora caninum* (*N. caninum*) strain NC-1 was extracted from tachyzoites cultured *in vitro*. The DNA of *Sarcocystis miescheriana* (*S. miescheriana*) was extracted from porcine tissue samples collected at a pig farm in Shangqiu City, Henan Province. The DNA of *Theileria luwenshuni* (*T. luwenshuni*), *Babesia bovis* (*B. bovis*), *Anaplasma phagocytophilum* (*A. phagocytophilum*), *Cyclospora cayetanensis* (*C. cayetanensis*), *C. parvum*, *G. duodenalis*, *Enterocytozoon bieneusi* (*E. bieneusi*), and *Eimeria* spp. of swine were stored in our laboratory. The DNA of *T. luwenshuni* and *B. bovis* was extracted from blood samples collected from Boer goats on a farm in Luoyang City, Henan Province. The DNA of *A. phagocytophilum* was extracted from blood samples collected from a dairy cow on a farm in Pingdingshan City, Henan Province. The DNA of *C. cayetanensis* was extracted from a fecal sample collected from a human patient at a hospital in Zhengzhou City, Henan Province. The DNA of *C. parvum*, *G. duodenalis,* and *E. bieneusi* was extracted from fecal samples collected from dairy cattle on a farm in Zhengzhou City, Henan Province. The DNA of *Eimeria* spp. (swine) was extracted from a fecal sample collected from a pig on a farm in Xuzhou City, Jiangsu Province. Information on 14 clinical samples tissue and six clinical blood samples used in this study is provided in Table S1. The negative tissue samples used in the sensitivity test were obtained from the healthy spleen tissues of euthanized SPF healthy Kunming mice known to be negative for *T. gondii*. The spleen tissue samples were immediately stored at −80°C upon collection and kept frozen until use.

### *T. gondii* pure culture and DNA extraction

After centrifuging the rejuvenated RH strain of *T. gondii*, we resuspended the sample in 0.2 mL of Dulbecco's Modified Eagle Medium cell culture medium supplemented with 10% heat-inactivated fetal bovine serum. The culture was then added to cell culture flasks containing Vero cells and incubated at 37°C under 5% $CO_2$. Once the peak release of tachyzoites was achieved, the culture supernatant was discarded, followed by the collection of the tachyzoites. The collected tachyzoites were suspended in phosphate-buffered saline and counted using an Xb-k-25 Hemocytometer. After counting was completed, they were added to negative blood and tissue samples, from which genomic DNA was extracted using a Blood/Cell/Tissue Genomic DNA Extraction Kit manufactured by TIANGEN (Beijing, China). A series of 10-fold dilutions was then prepared. To determine the limit of detection (LOD), each concentration of tissue-spiked DNA and blood-spiked DNA underwent replicate testing ($n = 3$).

### Design and synthesis of crRNA

The glycerol-3-phosphate dehydrogenase (B1) gene, present in 30–35 copies per genome of *T. gondii*, represents a highly conserved, multicopy sequence frequently used in PCR assays for the detection of *T. gondii* (27, 28). We selected this gene as the target sequence for designing the crRNA. Sequences of the B1 gene from various strains of *T. gondii* were downloaded from GenBank and aligned using ClustalX 2.1 (http://www.clustal.org/clustal2/). A highly conserved 24-nucleotide (nt) sequence immediately adjacent to a T nt-rich protospacer adjacent motif was selected as the target sequence. The target sequence, along with the T7 promoter sequence, was then synthesized (Table S2). A BLAST analysis was performed on the crRNA to verify the specificity of the synthesis. The two single-stranded RNAs, crRNA-F and crRNA-R, were synthesized by Sangon Biotech (Shanghai, China) and then annealed to produce dsDNA templates. The crRNA

was transcribed using the HiScribe T7 High Yield RNA Synthesis Kit from New England Biolabs (Ipswich, MA, USA). After transcription, the crRNA was treated with Recombinant DNase I from Takara (Dalian, China) and purified using NucAway Spin Columns (Thermo Fisher Scientific, Waltham, MA, USA). Finally, the concentration of the purified crRNA was determined using a NanoDrop One spectrophotometer (Thermo Fisher Scientific, Waltham, MA, USA).

## Recombinase polymerase amplification assay

Based on the sequence of the B1 gene in GenBank (accession number: MN275920), five pairs of RPA primers targeting the B1 gene were designed using the Primer-BLAST tool available online from the National Center for Biotechnology Information (Table S2). A TwistAmp Basic kit (TwistDx Ltd., Hertfordshire, UK) was used for RPA to amplify the target sequence of the B1 gene, in accordance with the manufacturer's instructions. In brief, a 50 µL reaction mixture was prepared containing 29.5 µL of rehydration buffer, 500 nM of each primer, 5 µL of extracted DNA, 2.5 µL of 280 nM magnesium acetate (MgOAc), sterile nuclease-free water, and the TwistAmp reaction pellet. The mixture was incubated at 37°C for 30 min. All components except MgOAc were added to the bottom of the reaction tube, while MgOAc was added to the lid of the tube. The tube was centrifuged to mix the MgOAc with the solution, thereby initiating the RPA reaction.

## FnCas12a/crRNA trans-cleavage assay

An FnCas12a *trans*-cleavage assay was performed as previously described (26). Fluorescence signals were monitored in real time using the qTOWER3G qPCR system (Analytik Jena, Germany). The system recorded fluorescence values at 30 s intervals, maintaining a constant temperature of 37°C for 25 min, with an excitation wavelength of 515 nm and an emission wavelength of 545 nm. For the LFS assay, the FnCas12a *trans*-cleavage assay was conducted following the same protocol. Subsequently, 30 µL of sterile nuclease-free water was added to the reaction mixture and mixed thoroughly. The LFS (WarBio, Nanjing, China) was then inserted into the reaction tube, and the results were visualized at room temperature within 7–10 min.

## Preparation of target DNA of *T. gondii* at a known concentration

For the sensitivity test, a DNA fragment of the *T. gondii* B1 gene was prepared at a known concentration through PCR amplification, using the primers B1-Tg1 and B1-Tg2 (Table S2). PCR amplifications based on the B1 gene were conducted as in a previous study (29). After that, the amplified DNA fragment was purified via gel extraction using a SanPrep Column DNA Gel Extraction Kit sourced from Sangon Biotech (Shanghai, China), according to the manufacturer's guidelines. The purified *T. gondii* B1 gene DNA fragment's final concentration was determined using a NanoDrop One spectrophotometer (Thermo Fisher Scientific Inc., Waltham, MA, USA). This was followed by a series of 10-fold dilutions. To determine the LOD, each concentration of target DNA underwent replicate testing ($n = 3$).

## Nested PCR amplification of the *T. gondii* B1 gene

The REPORT detection method for *T. gondii* detection was compared with the nested PCR method based on the B1 gene for *T. gondii* detection. We selected 14 tissue samples and six blood samples for simultaneous testing using both methods. PCR amplification based on the B1 gene employing the primers B1-Tg1, B1-Tg2, B1-Tg3, and B1-Tg4 (Table S2) was conducted as described in a previous study (29).

## Statistical analysis

Statistical analyses were performed using GraphPad Prism 8. Data are presented as mean ± SD. $P < 0.001$ indicated a statistically significant difference.

## RESULTS

### Design and preparation of crRNA

The B1 gene sequences of different strains of *T. gondii* were downloaded and aligned. A highly conserved 20 nt sequence was selected as the target sequence, and the crRNA sequence was also determined (Table S2 and Fig. S1). The BLAST result for the crRNA sequence showed that it did not cross-react with other species and had good specificity (Fig. S2). The crRNA was successfully prepared by annealing synthesized single-strand crDNA oligonucleotides followed by *in vitro* transcription and purification. The concentration of this crRNA-Tg was measured as 1,695.71 ng/µL using a NanoDrop One spectrophotometer (Fig. S3 and S4).

### Recombinase polymerase amplification primer screening

Five pairs of RPA primers were individually subjected to RPA amplification, and the RPA products were subsequently analyzed using the FnCas12a/crRNA trans-cleavage assay. The primers F102-132 and R295-326, when combined with crRNA-Tg, exhibited the highest fluorescence value and were selected for the present study (Fig. 1a). In the REPORT-based fluorescence detection assay, the activated FnCas12a enzyme effectively cleaved the HEX-12N-BHQ1 reporter, enabling the visual observation of a fluorescent signal under UV light; this signal was notably absent in the negative reaction (Fig. 1b). In the REPORT-based LFS detection assay, a test line appeared on the strip within 10 min after the FAM-12N-Biotin reporter was cleaved by the activated FnCas12a; however, the test line did not appear in tests with negative samples (Fig. 1c).

### Sensitivity of the REPORT-based detection method for *T. gondii* (target DNA)

A fragment of the *T. gondii* B1 gene was prepared using PCR amplification (Fig. S5). The concentration of the purified DNA fragment was 13.2 ng/µL, corresponding to $2.2 \times 10^{10}$ copies/µL. The prepared target DNA subsequently underwent a series of dilutions, ranging from $2.2 \times 10^9$ copies/µL down to 2.2 copies/µL. The fluorescence detection method demonstrated that when the sample concentration was greater than or equal to 2.2 copies/µL, there was a robust fluorescent signal recorded using the qTOWER3G qPCR system that was visible to the naked eye under blue light and that was significantly different from the low-concentration samples and the negative control (Fig. 2a through c). In the REPORT-based LFS detection method, the same results were observed, and the

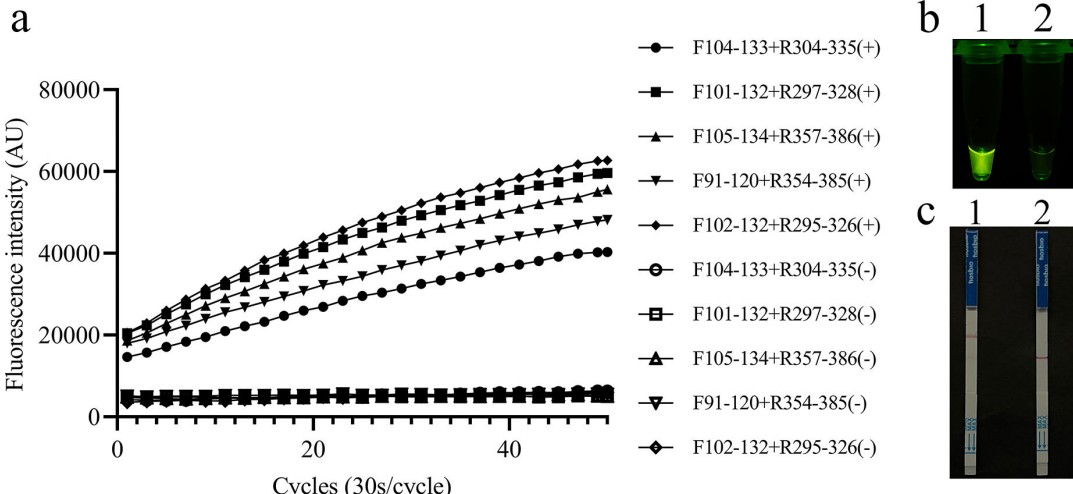

**FIG 1** Screening for optimal primer pairs for RPA. The primer pairs listed in Table S1 were also tested. (a) Primer pair F104-133 and R304-335 for *T. gondii* was the best. (b) A clear fluorescent signal can be observed under UV light by the naked eye. 1: positive result; 2: negative result. (c) A clear test line can be observed in the LFS of assemblage A. 1: positive result, 2: negative result.

test line only appeared on the test strip when the sample concentration was greater than or equal to 2.2 copies/µL (Fig. 2d). According to these results, the LOD of the REPORT-based nucleic acid detection of *T. gondii* was 2.2 copies/µL of target DNA.

## Sensitivity of the REPORT-based detection of *T. gondii* (genomic DNA)

To evaluate the sensitivity of the REPORT-based method in clinical applications, we added a known concentration of tachyzoites to *T. gondii*-negative tissue and blood samples and extracted the genomic DNA (Fig. S6). The concentration of tachyzoites in the tissue sample was $3.1 \times 10^6$ tachyzoites per gram of tissue (tachyzoites/g). After extracting DNA from the tissue sample, the DNA was serially diluted to $3.1 \times 10^5$ to $3.1 \times 10^{-1}$ tachyzoites/g. In the REPORT-based detection, samples with 3.1 tachyzoites/g or higher exhibited stronger fluorescent signals than samples with lower concentrations and the negative control (Fig. 3a through c). Comparable results were observed in the REPORT-based LFS detection (Fig. 3d). The concentration of tachyzoites in the blood sample was $5 \times 10^6$ tachyzoites per milliliter of blood (tachyzoites/mL). After extraction from the sample, the DNA was serially diluted to five tachyzoites/mL. In the REPORT-based fluorescence detection and LFS detection, samples with five tachyzoites/mL or higher showed robust fluorescent signals and a clear test line that distinguished those samples from the negative control (Fig. 3e through h). The above results show that the LOD values for the REPORT-based tissue and blood sample detection of *T. gondii* were 3.1 tachyzoites/g and five tachyzoites/mL, respectively.

## Specificity of the REPORT-based detection of *T. gondii*

We verified the specificity of the REPORT-based detection using genomic DNA from several prevalent tissue, blood, or intestinal parasites, including *N. caninum*, *S. miescheriana*, *T. luwenshuni*, *B. bovis*, *A. phagocytophilum*, *C. cayetanensis*, *C. parvum*, *G. duodenalis*, *E. bieneusi*, and *Eimeria* spp. of swine. Using the qTOWER3G qPCR system, a strong fluorescence intensity was detected that was significantly different from those for the

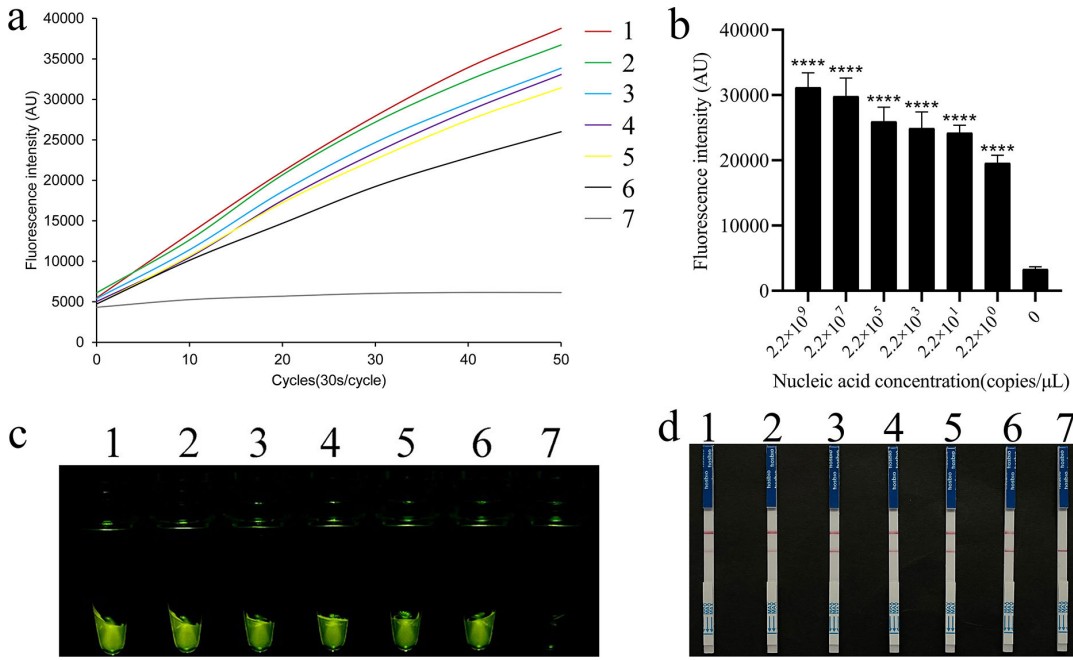

**FIG 2** Sensitivity of the REPORT-based detection of target DNA. The sensitivity of the REPORT-based *T. gondii* detection method was assessed using real-time fluorescent signals (a), quantitative analysis (b) (****$P < 0.0001$; the bars represent the means ± SEMs), and visible green fluorescence (c). (d) Sensitivity of the REPORT-based LFS detection of *T. gondii* for various concentrations of purified target DNA (1–7: $2.2 \times 10^9$, $2.2 \times 10^7$, $2.2 \times 10^5$, $2.2 \times 10^3$, $2.2 \times 10^1$, $2.2 \times 10^0$, 0 CFU/mL, respectively).

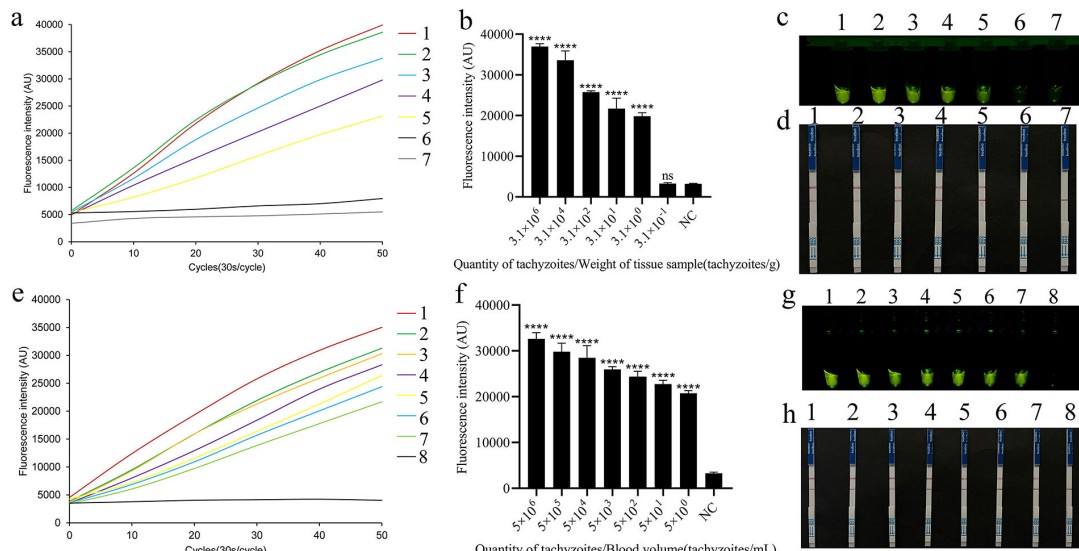

**FIG 3** Sensitivity of REPORT-based detection of tissue and blood samples. The sensitivity of the REPORT-based *T. gondii* detection method in tissue samples was assessed using real-time fluorescence signals (a), quantitative analysis (b) (****$P$ < 0.0001; the bars represent the means ± SEMs), and visible green fluorescence (c). (d) Sensitivity test of the REPORT-based LFS detection of *T. gondii* for various tachyzoites in tissue samples. 1–7: $3.1 \times 10^6$, $3.1 \times 10^4$, $3.1 \times 10^2$, $3.1 \times 10^1$, $3.1 \times 10^0$, $3.1 \times 10^{-1}$, 0 tachyzoites/g, respectively. The sensitivity of the REPORT-based *T. gondii* detection method in blood samples was assessed using real-time fluorescence signals (e), quantitative analysis (f) (****$P$ < 0.0001; the bars represent the means ± SEMs), and visible green fluorescence (g). (h) Sensitivity test of the REPORT-based LFS detection of *T. gondii* for various tachyzoites in blood sample, 1–8: $5 \times 10^6$, $5 \times 10^5$, $5 \times 10^4$, $5 \times 10^3$, $5 \times 10^2$, $5 \times 10^1$, $5 \times 10^0$, 0 tachyzoites/mL, respectively.

other parasites and the negative control in samples positive for *T. gondii* (Fig. 4a and b). Only when *T. gondii* DNA was used as the template could a strong fluorescence under UV light and a clear test line on LFS be visually observed (Fig. 4c).

## Performance of the REPORT-based detection of *T. gondii* on clinical samples

To assess the performance of the REPORT-based detection method for *T. gondii* in clinical application, a comparative analysis was conducted using a total of 14 tissue samples and six blood samples. The samples were tested using both the REPORT method and nested PCR targeting the B1 gene sequence. The nested PCR successfully amplified specific fragments of approximately 530 bp in 11 tissue samples and three blood samples (Fig. S7 and S8). It is worth noting that the qTOWER3G qPCR system detected clear fluorescent signals in the same samples, and these were also visually observable. Meanwhile, no such signals were present in the PCR-negative samples, which indicated a perfect concordance with PCR-based nucleotide sequencing results (Fig. 5a, b, 6a and b). Furthermore, the LFS detection based on the REPORT system demonstrated complete agreement with the conventional PCR method, achieving a 100% concordance rate (Fig. 5c and 6c). Using nested PCR as the reference standard (Table S3), the REPORT method achieved 100% sensitivity (14/14), 100% specificity (6/6), and 100% overall accuracy (20/20) for both fluorescence and LFS readouts.

## DISCUSSION

There is currently no effective drug for treating *T. gondii* infection. Therefore, to control and prevent the spread of the disease, it is necessary to develop a rapid and accurate early diagnostic method. Serological testing is the most commonly used method for detecting *T. gondii*, but it has a long window period and may result in missed detection (12). Traditional amplification methods, such as PCR and qPCR, are time-consuming (requiring 2 h or more) and rely on expensive equipment and well-trained operators. In a previous work, we reported a diagnostic method termed REPORT for rapid detection of

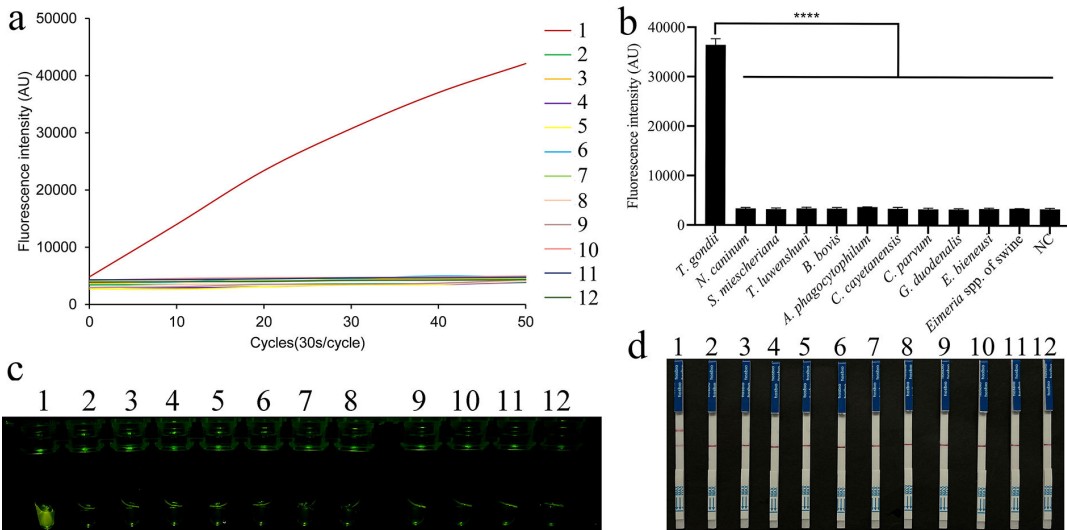

**FIG 4** Specificity of the REPORT-based detection. Genomic DNA of *T. gondii*, *N. caninum*, *S. miescheriana*, *T. luwenshuni*, *B. bovis*, *A. phagocytophilum*, *C. cayetanensis*, *C. parvum*, *G. duodenalis*, *E. bieneusi*, and *Eimeria* spp. of swine (1–11) was included, NC: Negative sample. (a and b) Specificity test results of REPORT-based fluorescence assays were recorded using the real-time fluorescence signals (a), quantitative analysis (b) (****$P$ < 0.0001; the bars represent the means ± SEMs), and visibly observed under UV light (c). Only samples of *T. gondii* exhibited strong fluorescence signals (d). Specificity test of the REPORT-based LFS detection assay. A clear test line was observed only on the LFS that contained *T. gondii* DNA.

*G. duodenalis* nucleic acid based on CRISPR/Cas12a (26). In this study, we expanded the REPORT system and established an end-point detection method for *T. gondii* targeting the B1 gene.

The selection of an appropriate nucleic acid amplification technique has a crucial role in clinical diagnosis (30). RPA primer design is relatively simple, and the RPA assay is both sensitive and accurate. Importantly, the amplification reaction can be performed at 37°C, requiring only 20–30 min to complete. In this study, the RPA assay achieved substantial amplification of potential target nucleic acid fragments (target DNA) present in the samples. The amplification resulted in an increase of more than $10^{12}$ (31). For the REPORT-based detection, the sensitivity test results showed that the LOD of the target

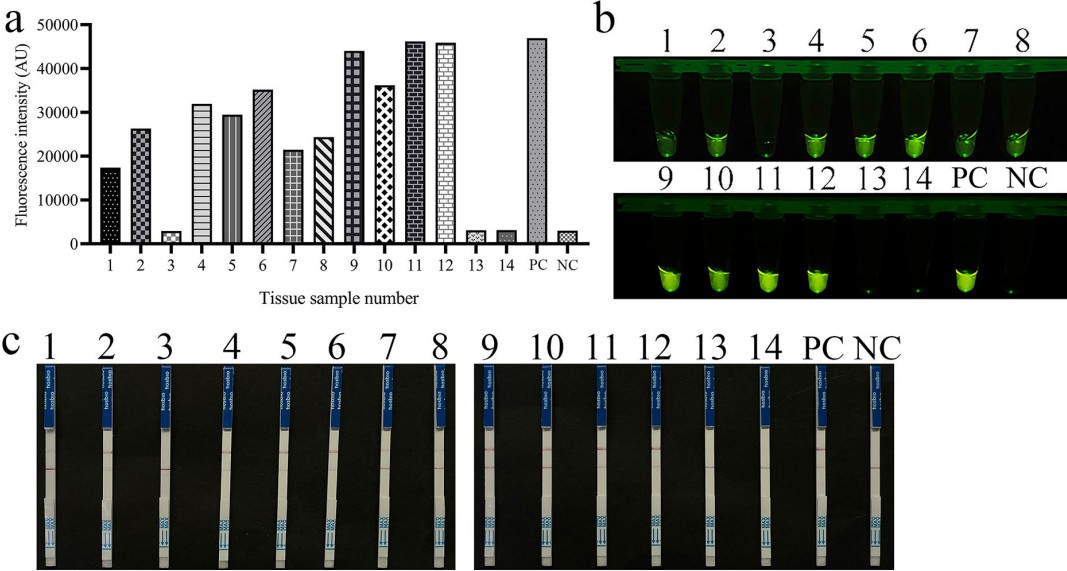

**FIG 5** REPORT-based detection of *T. gondii* on clinical tissue samples. Fourteen clinical tissue samples were tested using REPORT-based fluorescence (a and b) and LFS (c) detection methods. 1–14: clinical tissue samples; PC: positive control; NC: negative control.

DNA was 2.2 copies/µL. When analyzing tissue and blood samples, the respective LOD values were 3.1 tachyzoites/g and five tachyzoites/mL. These results further validate the exceptional amplification efficiency of RPA. When one Cas12a is activated by target DNA, it can rapidly cleave a large number of ssDNA probes, a property that may contribute to the high sensitivity of the REPORT-based method. Previous studies have shown that Cas12a cleavage not only serves as a signal transducer, generating a strong fluorescent signal and producing a detection line on the test strip but is also involved in the signal amplification process (32).

The BLAST results for crRNA-Tg demonstrated that it had 100% similarity exclusively with different strains of *T. gondii*, suggesting that there was no cross-reaction with other species. The high specificity of the REPORT-based detection method was further confirmed in the tests involving other common parasites. The specificity test, which included several common tissue, blood, and intestinal parasites, further confirmed the specificity of the REPORT-based detection method for *T. gondii*. Cas12a is activated when the spacer region of the crRNA (or gRNA) forms a reverse complementary alignment with the target protospacer sequence, endowing it with enzymatic cutting properties, with a tolerance for only one or two discontinuous nucleotide mismatches in the reverse complementarity match (33). This implies that the DNA fragment to be tested must have nearly 100% homology with the crRNA; this is also a prerequisite for the specificity of the REPORT method. Therefore, the fluorescent signal visually observed or the test line shown on the LFS is only apparent if there are orthologs present in the test samples. Only when the amplified target DNA is homologous with the crRNA can Cas12a cleave the ssDNA probe, resulting in a visually observable fluorescent signal or a test line appearing on the LFS.

Using REPORT as an end-point detection method in clinical tissue and blood samples, the results can be obtained in approximately 1 h, which is significantly less than the time needed for the nested PCR method. This test can be performed at a constant temperature of 37°C without the need for expensive thermal cyclers. The results can be determined by observing either the presence of strong fluorescence under a blue light device or the appearance of a detection line on the test strip, eliminating the requirement for professional technical personnel and enabling on-site detection. We compared the detection efficiency of the REPORT system and nested PCR in clinical tissue and blood samples. There was 100% consistency between the two methods, indicating that the REPORT system can be applied in POC detection.

In addition to the aforementioned advantages, the REPORT method has several limitations. This REPORT method can identify whether the host is infected with *T. gondii*; however, it cannot determine which strain of *T. gondii* is causing the infection. Owing to the presence of differential gene expression among *T. gondii* strains leading to

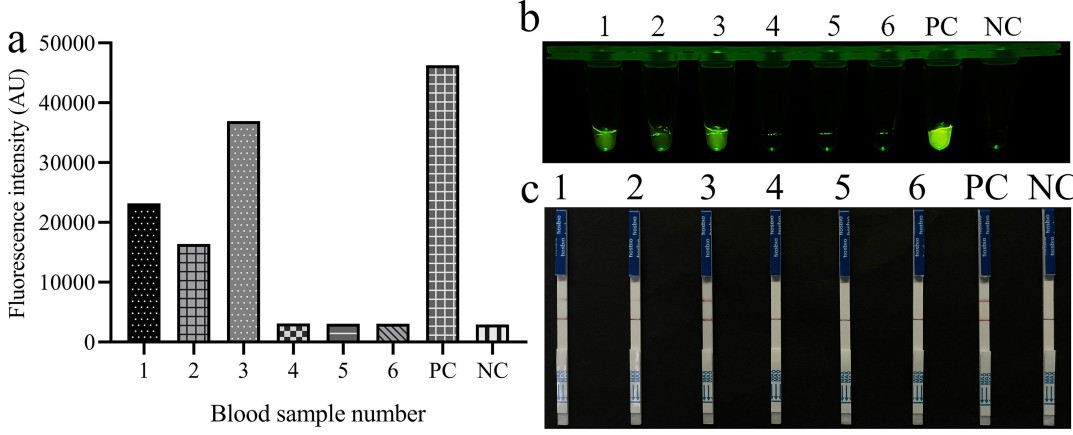

**FIG 6** REPORT-based detection of *T. gondii* on clinical blood samples. Six clinical blood samples were tested using REPORT-based fluorescence (a and b) and LFS (c) detection methods. 1–6: clinical blood samples; PC: positive control; NC: negative control.

variation in virulence and pathogenicity, subsequent research should focus on designing species-specific sgRNAs that would be particularly beneficial for identifying and typing *T. gondii* genes.

## Conclusion

The previously established REPORT method, which has been extended for the detection of *T. gondii* nucleic acid, is characterized by high specificity and sensitivity. It also shows excellent performance for detecting *T. gondii* in clinical tissue and blood samples. This REPORT-based detection method for *T. gondii* has a significant potential for POC detection in resource-scarce areas and is expected to replace the existing methods.

## ACKNOWLEDGMENTS

The authors thank Yuhang Zhang and Yafei Chang for their technical support, Professor Liu Qun for providing the T. gondii tachyzoites (RH strain) and N. caninum DNA, Professor Li Yongtao for supplying the Vero cells, andd Professor Yan Wenchao for providing the S. miescheriana DNA. The authors also extend their appreciation to Professor Yang Yurong for contributing the tissue samples (Nos. 1–10) and blood samples (Nos. 15–17).

This work was supported by the Key Research and Development Special Project of Henan Province of China (231111111500) and the National Key Research and Development Program of China (2023YFD1801200,2022YFD1800200).

Y.W.: data curation; writing—original draft; visualization; and methodology. Z.Q.: conceptualization and data curation. Q.W.: writing—review and editing and visualization. Y.Y.: supervision and conceptualization. C.G.: formal analysis and visualization. F.Y.: methodology; writing—review and editing; and validation. Y.W.: software and writing—review and editing. L.X.Z.: conceptualization, supervision, grant funding, project administration, and writing—review and editing.

## AUTHOR AFFILIATIONS

[1]College of Veterinary Medicine, Henan Agricultural University, Zhengzhou, Henan, People's Republic of China

[2]International Joint Research Laboratory for Zoonotic Diseases of Henan, Zhengzhou, People's Republic of China

[3]Key Laboratory of Quality and Safety Control of Poultry Products, Ministry of Agriculture and Rural Affairs, Zhengzhou, Henan, People's Republic of China

[4]College of Animal Science, Tarim University, Alar, Xinjiang, People's Republic of China

## AUTHOR ORCIDs

Yurong Yang http://orcid.org/0000-0001-8971-4648
Long xian Zhang http://orcid.org/0000-0001-9310-1975

## FUNDING

| Funder | Grant(s) | Author(s) |
| --- | --- | --- |
| Key Research and Development Special Project of Henan Province of China | 231111111500 | Long xian Zhang |
| National Basic Research Program of China | 2022YFD1800200, 2023YFD1801200 | Long xian Zhang |

## AUTHOR CONTRIBUTIONS

Yilin Wang, Data curation, Methodology, Visualization | Ziyang Qin, Conceptualization, Data curation | Qinglin Wang, Visualization | Yurong Yang, Conceptualization, Supervision | Chunhao Gu, Formal analysis, Visualization | Fuchang Yu, Methodology, Validation

| Yayun Wu, Software | Long xian Zhang, Conceptualization, Project administration, Supervision

## DATA AVAILABILITY

The data supporting this study's findings are available on request from the corresponding author, Long xian Zhang: zhanglx8999@henau.edu.cn.

## ETHICS APPROVAL

All the research procedures used in this study were approved by the Institutional Review Board of Henan Agricultural University (approval no. IRB-HENAU-20190820-02). The use of positive human samples in this study complied with the 1975 Declaration of Helsinki, as revised in 2013.

## ADDITIONAL FILES

The following material is available online.

### Supplemental Material

**Supplemental material (Spectrum01550-25-s0001.docx).** Supplemental figures and tables.

### Open Peer Review

**PEER REVIEW HISTORY (review-history.pdf).** An accounting of the reviewer comments and feedback.

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
