## [Reviewer comments · Microbiology Spectrum]

Microbiology Spectrum

An RPA-CRISPR/Cas12a-based rapid and sensitive nucleic acid method for detection of *Toxoplasma gondii* in tissue and blood samples

Yilin Wang, Ziyang Qin, Qinglin Wang, yurong Yang, Chunhao Gu, Fuchang Yu, Yayun Wu, and Longxian Zhang

Corresponding Author(s): Longxian Zhang, Henan Agricultural University

Review Timeline:

Submission Date:	July 22, 2025
Editorial Decision:	November 10, 2025
Revision Received:	November 11, 2025
Accepted:	November 14, 2025

Editor: Rosemary She

Reviewer(s): Disclosure of reviewer identity is with reference to reviewer comments included in decision letter(s). The following individuals involved in review of your submission have agreed to reveal their identity: Saleem Khteer Al-hadrawy (Reviewer #1)

Transaction Report:

DOI: <https://doi.org/10.1128/spectrum.01550-25>

Re: Spectrum01550-25 (An RPA-CRISPR/Cas12a-based rapid and sensitive nucleic acid method for detection of *Toxoplasma gondii* in tissue and blood samples)

Dear Dr. Longxian Zhang:

Thank you for the privilege of reviewing your work. Below you will find my comments, instructions from the Spectrum editorial office, and the reviewer comments.

I am pleased to inform you that your manuscript has been editorially accepted for publication. However, there are a few additional questions in the submission form that need to be answered before the final decision. Once these are completed, please return your submission so that I can move your paper forward to acceptance.

Sincerely,
Rosemary She
Editor
Microbiology Spectrum

Reviewer #1 (Comments for the Author):

The study is characterized by accuracy and scientific laboratory methodology in applying the parasites that represent a study and which represent a health problem in many regions of the world.

Re: Spectrum01550-25R1 (An RPA-CRISPR/Cas12a-based rapid and sensitive nucleic acid method for detection of *Toxoplasma gondii* in tissue and blood samples)

Dear Dr. Longxian Zhang:

Your manuscript has been accepted, and I am forwarding it to the ASM production staff for publication. Your paper will first be checked to make sure all elements meet the technical requirements. ASM staff will contact you if anything needs to be revised before copyediting and production can begin. Otherwise, you will be notified when your proofs are ready to be viewed.

Sincerely,
Rosemary She
Editor
Microbiology Spectrum